# Strategic Turnaround in the Paper Industry: A New Model for the Procurement of Recycled Paper

Golub V. Marković [1,*] and Marko M. Mihić [2,*]

1   Bunzl & Biach Gesellschaft m.b.H., Steinheilgasse 5, 1210 Vienna, Austria
2   Faculty of Organizational Sciences, University of Belgrade, 154 Jove Ilića, 11000 Belgrade, Serbia
*   Correspondence: markovic@bunzl-biach.at (G.V.M.); marko.mihic@fon.bg.ac.rs (M.M.M.)

**Abstract:** In recent decades, the paper industry has undergone many changes, making this industry more lucrative, "greener", and thus more attractive. Today, recycled paper is a key raw material for paper production. However, this intense growth has also increased the number of market players, making competition more intense and dynamic and causing frequent turmoil. In such an environment, planning procurement and forecasting the price of recycled paper is a big challenge, even for highly experienced procurement managers. In addition, paper production itself is a dynamic process that requires all key grades of recycled paper to be available at all times. Accordingly, managing the optimal level of recycled paper stocks is also a difficult task faced by the procurement unit. The goal of this paper is to address these challenges by developing a new model for the sustainable procurement of recycled paper in the paper industry with the help of the principles of strategic management. Specifically, we aim to enhance our understanding of the factors driving the complexity of recycled paper procurement management. To this end, we conducted a comparative case study in four European companies. To build cases, we collected secondary data on the sampled companies as well as primary data from interviews with top executives at these companies. On the basis of the results from the comparative case study, we propose a new model for the procurement of recycled paper that helps increase the accuracy of forecasting price trends and by extension overall procurement performance. In a nutshell, this paper seeks to improve procurement processes by reducing the complexity of the enterprise procurement unit as well as by offering guidelines for the maintenance of optimal stock levels of recycled paper.

**Keywords:** recycled paper procurement model; strategic management; organization; paper industry

## 1. Introduction

### 1.1. The Importance of Procurement Management in the Paper Industry

Procurement management is a vital management function. In transactions between firms, procurement is important for both buyers and sellers because it represents the intersection of supply and demand. Accordingly, the scope of the procurement function includes both internal and external elements of a firm's supply chain. A well-performing procurement method is thus necessary for both buyers and suppliers as they must effectively negotiate in the market and agree on the prices that are acceptable to both parties. This is particularly important for firms in the paper industry, which must combine dynamic market information with knowledge about raw materials (i.e., recycled paper) in order to make informed procurement decisions and perform well.

The procurement function creates a critical link between the short term profit orientation of a firm and its core business strategy as it relates to firm sustainability [1]. Procurement plays a key role in sustainability as it encompasses practices beyond organizations' boundaries, incorporating their entire supply chains [2,3]. Sustainable procurement is therefore increasingly on the agenda of top executives seeking to demonstrate corporate

social responsibility in their supply chains [4]. In the paper industry in particular, the sustainable procurement of recycled paper is an example of such efforts [5].

Procurement involves a wide range of activities such as supplier selection, price negotiation, contract management, transaction management, and supplier performance management [6]. With the increasing availability of outsourcing options, procurement functions require not only cross-functional integration but also a high level of interorganizational integration [1,7]. To that end, professionals in the field of raw material management often say that "the most expensive raw material for the factory is the one that is not in stock".

The specificity of raw material procurement depends on the nature of the company's industry [8]. As a centuries-old industry, the paper industry has made significant changes in its recent history. With the emergence of modern business and digitalization, the strengthening awareness of ecology, worldwide economic growth, demographic development, and the spread of literacy, enhancing industrial production was followed by the introduction of circular economy principles, meaning that this industry is going through an intense transition period. One of the essential changes that happened along the way is the transition from cellulose to recycled paper in the production process. A new modification in the recipe, in addition to environmental benefits, led to the expansion of the global recycled paper market and gave this industry a "green" image.

In this industry, firms supply the paper mills with the basic raw material, i.e., recycled paper (renewable fibers) and to a lesser extent cellulose (unprocessed fibers). The complexity of the production process is accompanied by extremely volatile prices and a dynamic environment [9–11]. The procurement of raw materials and related management activities are two key concerns of any production organization [12]. Perhaps because of the strategic importance of procurement, firms in the paper industry rarely outsource this functional unit.

The procurement of recycled paper in the paper industry is characterized by the following key characteristics [13]:

1. It involves large quantities of raw material;
2. The continuity of procurement is vital—it is a continuous process;
3. The procurement of raw material is international (it involves global markets);
4. Procurement managers must be highly trained specialists.

The procurement of recovered paper in the paper industry requires a complex organizational structure because it takes place mainly through a collection network distributed in the form of divisions (branches) by markets (regions and countries). In the existing procurement models of firms, there are frequent communication "noises", overlapping competencies, and employee dissatisfaction, which call into question the efficiency of the recycled paper procurement unit. In addition, the existing procurement models tend to lead to significant deviations from the annual adopted plans of recycled paper procurement, which is caused by unstable production, pronounced market imbalances, changes in global market flows, and other factors [14]. Furthermore, for decades the European Union (EU) has been encouraging paper producers to comply with the principles of the circular economy, which directly affects the positive trends in this industry. The European Declaration of 2000 defined goals for the continuous growth of the recycling rate of paper and other packaging waste, which was a strong impetus for the paper industry.

Based on a comprehensive literature review on this topic, we found that it has not received much attention in recent years, as is evidenced by the lack of studies at the intersection of procurement and strategic management in the paper industry. The challenges of modern procurement in the paper industry described above, in combination with the lack of research on this topic, were the main motivators for the current research. Accordingly, we set out as the main research goal to investigate how principles of strategic management can be systematically implemented in the domain of the procurement of recycled paper as a means of improving overall procurement performance [15–17].

*1.2. The Limits of Current Procurement Management Models*

In today's business environment, the task of procurement function not only involves reducing costs and achieving a good value/money ratio but also achieving broader policy goals such as social, environmental, or other goals [18,19]. In other words, the procurement function in firms rarely serves only one purpose. A solid working relationship between purchasing and procurement organizations has been highlighted to reduce uncertainty and to encourage innovative supplier responses [20]. This applies to small family businesses and large corporations.

Purchasing activity is always related to corporate and business strategy. Therefore, it should be systematically planned, evaluated, implemented, and controlled to achieve the company's long-term goals [21,22]. In order for a firm to be able to compete successfully in today's uncertain environment, it should develop a sophisticated purchasing function that is integrated into the strategic management decision-making process. Three key factors appear to have created the opportunity and necessity for an increase in purchasing sophistication and integration into the strategic decision-making process. They are the higher level of competition, the dynamic supply environment of the firm, and the changing nature of the purchasing function [23].

Because these factors vary from product to product, there cannot be a single universal procurement strategy for all products [23,24]. To maximize its contribution to the overall performance of the company, according to Puler, procurement must establish the following two overarching goals [25]:

(1) Ensure an economically stable supply through the procurement of raw materials, goods, other supplies, and services for the company to function smoothly.
(2) Contribute to profits by effectively controlling the company's total operations cost. Although managers often give priority to the first goal, both are equally important.

The procurement function implies a focus on both goals because effective control of procurement costs directly affects the company's bottom line [26]. After fulfilling the first point and ensuring stable procurement, it is necessary to handle the second point wisely and fight for the best possible result in terms of procurement costs. This is especially the case in the paper industry, where the required quantity of recycled paper is certainly the main priority. Additionally, it is necessary to consider the purchase prices, i.e., the direct costs of procurement, which affects the profitability of the entire organization.

Given the growing complexity of raw material procurement processes and the increasing global orientation towards procurement activities, comprehensive and systematic procurement market research is essential for assessing market risks and opportunities and for optimal product, material, raw material, and procurement services. For this reason, procurement market research is one of the key tasks carried out by procurement managers in the paper industry [27–29]. Whether it is about strategic or tactical plans, Burt and Pinkerton point out that the procurement planning process always involves four phases:

(1) Current situation analysis (also called an audit)—a review of the procurement system or a diagnostic phase; at this stage, it is necessary to analyze the differences between the set goals and the results so far.
(2) Goals phase—from the procurement aspect, it is important to define specific and clear goals that the entire organizational unit will understand and pursue together with the responsible manager.
(3) Plan development phase—based on these goals, responsible procurement managers begin to develop a plan by using their experience to analyze reports and historical data related to the specifics of the procurement they lead; special focus is placed on the assumptions for the future of the business based on external conditions.
(4) Implementation–monitoring–audit phase—after creating a procurement plan, which often contains sub-plans, the plan is submitted to the board of directors for approval or correction.

Raw material procurement management today is an even greater challenge for companies in an increasingly dynamic environment and with uncertain global market competition. This is especially the case in the paper industry. Based on our research into the paper industry, we found that there are shortcomings in the existing procurement systems of recycled paper. Accordingly, we posed the following research questions:

(1) What factors influence the management of recycled paper procurement in firms in the paper industry?
(2) What are the typical problems faced by the organizational unit in terms of recycled paper procurement?
(3) What are the shortcomings of the existing management models for recycled paper procurement?

*1.3. Research Problem and Goals*

We identified several issues with the current management models for the procurement of recycled paper. First, strategic management principles are rarely applied in this domain. Second, firms often face the issue of forecasting price trends precisely and cannot maintain the optimal level of recycled paper stocks. Third, there is the problem of the complexity of the organizational unit responsible for recycled paper procurement management. Finally, firms in the paper industry often suffer from communication noise, which leads to a decrease in the efficiency of the recycled paper procurement unit.

Our premise in this research is that these issues can be resolved, at least in part, by the systematic application of strategic management in the procurement of recycled paper. To examine this proposition, we conduct a literature review and apply multiple case study methodologies encompassing four firms in the paper industry to propose a novel strategic management model for the procurement of recycled paper. Specifically, based on the integration of the theoretical insights from the review of selected strategic management models and the inputs obtained from case studies, we propose that the application of this new model can contribute to the overall performance of the company.

To address our research problem and build four cases studies, we conducted a comprehensive survey and semi-structured interviews, which we combined with archival data from our sampled companies in the paper industry as well as official reports from public institutions. The novelty of this research lies in the semi-structured interviews we conducted with 14 top executives (experts in the paper industry). While the survey aimed at thoroughly covering all topics related to the issues of this research, the follow-up interviews were used to obtain more detailed information regarding the main phenomena of interest.

With such an empirical approach, we aimed at identifying the shortcomings of the current procurement management models used by firms in the paper industry. We then examined the possibility of introducing strategic principles into the procurement system to improve the procurement process. This research was conducted under the assumption that the existing models cannot be simply applied by firms in the paper industry because those models do not cover all the specifics of the industry.

We therefore aimed at creating a new model that brings together certain elements of the existing models while applying strategic principles on the basis of which new elements are incorporated into our new model. Depending upon the size, the complexity of the offer, the staff employed, prices, and other factors that affect procurement organizations, procurement systems are run differently. Our strategic model of a procurement system accounts for these differences, proposing more adequate procurement management that is well aligned with the organization's overall business policy and operations.

## 2. Literature Review

*2.1. Implementation of Strategic Principles in Procurement Management*

According to Weigel, purchasing strategies should not be developed on the basis of conjecture and gut instinct [28]. Goals must always be defined on the basis of hard facts and figures. For that reason, it is important to collect extensive findings from internal

and external data at the outset in order to give them due consideration when strategies are developed. In this respect, internal data, resources, and organizational structures are as important as external market trends and competitive situations [28]. Guarnieri and Gomes examined whether strategic management principles can be implemented in public procurement. Based on the findings of these authors, it is possible to think strategically about procurement. Considering the concept of strategic procurement, they defined the following phases of public procurement: (1) planning and definition of objectives and criteria, (2) preparation of bidding notice, and (3) monitoring of the contract/purchasing agreement, which includes an analysis of the quality and level of the products or services provided [30]. Furthermore, Lee at al. investigated procurement strategies in a steel enterprise to determine the optimal procurement quantity to maximize profit through forward contracting and the spot market. Their approach to procurement strategies sees the procurement process along a time horizon from the buyer's perspective, with consideration of uncertain yields, stochastic demand, and dynamic spot market prices. The results of that case study indicate that a strategic approach enables buyers to achieve higher profits under volatile demand conditions [31].

### 2.1.1. Raw Material Procurement Specifics in Challenging Global Conditions

In general, raw material markets are influenced by the buyers and sellers' behavior around the world, which includes the work of large trading and brokerage organizations with a strong influence on price movements [32,33]. This general feature of the raw materials market largely coincides with the global market for recycled paper, i.e., renewable fibers. Therefore, the problem of raw material procurement in uncertain markets with fluctuating prices is common for buyers and consumers in most countries around the world. Some difficulties in raw material procurement management are inherent in market situations where suppliers have to operate, while others represent the response on the supplier side in some situations [32]. Kingsman and colleagues single out several key challenges that characterize the procurement management of firms in the paper industry.

### 2.1.2. Procurement Decision-Making under Conditions of High Uncertainty

According to Kingsman et al., procurement decisions must be made with great uncertainty about future market conditions, especially future price levels [33,34]. An industrial buyer, regardless of its size, cannot control prices, either now or in the future. They point out that the role of the purchaser is basically that of an observer who can only monitor how prices change and decide on a certain price at a certain point. Because of the raw material prices, international raw material markets are organized in such a way that they prevent any individual from influencing price flows. Although the required raw material can be purchased directly by the industrial supplier from the producer, the price will usually be determined by the market. The raw materials market can be influenced by many intermediaries acting on behalf of producers, consumers, speculators or on their own behalf [33,34]. Kingsman et al. also point out that the buyer cannot participate directly in price formation. Although the decisions the buyer makes in the end will certainly have a certain effect on prices, they will represent only a small part of the total consumer group. The situation in the global raw materials market is constantly changing. Although suppliers are aware that significant unexpected changes are possible, in many cases they cannot be predicted or anticipated. Such market dynamics are present in the paper industry, where decisions about the procurement of recycled paper are often made in conditions of high market uncertainty and involve various difficulties in predicting future industry trends.

### 2.1.3. Intense Fluctuations in Purchase Prices

Large fluctuations in purchase prices are also at play both in the long and short term. Consequently, uncertainty is further exacerbated as these events manifest themselves in large price changes. The renewable fiber market is a typical example of large price oscillations that occur over both short and long time intervals [9,35]. In addition, Bajpai and

Diesen emphasize that raw material markets act through the price mechanism as a means of rational supply of a given material among potential consumers. Prices depend more on the gap between supply and demand than on the absolute levels of any one factor. As a result, large price changes are needed to reduce consumption and reduce inventory shortages or, conversely, to increase consumption. Additionally, raw material prices from time to time become very sensitive to marginal changes in supply or demand [36]. Because people can form different views on the magnitude of these factors at any one time, prices can vary significantly even over the course of days as market interest varies between alternative positions [37].

### 2.1.4. Insufficiently Developed Raw Material Procurement Processes

Kingsman et al. explain that one of the reasons for the many difficulties that arise in this area in many companies is that the various stages in the process have not been formally analyzed and developed, resulting in a final detailed procurement decision. We point out that procurement experts have not paid enough attention to these problems. Economists have also done little to help procurement managers understand and deal with these issues. Additionally, formal training for the managing role in raw materials procurement is not sufficiently developed and processed.

Expectations of managers leading the raw material procurement sector are high and they are expected, among other things, to develop a "market sense" intuition about future price movements. Specific guidelines on how to do this are usually lacking or are not systematically processed. Therefore, predicting the raw materials price is often considered one primary problem that needs to be solved [32,38]. In such a setting, it often happens that all focus and efforts are based on more accurately predicting future prices while other aspects are neglected, such as research on other markets and suppliers, raw material substitutes, and similar activities in the form of trade options and insurance as alternative solutions [34,39]. Predicting future price trends of a given raw material is an indispensable continuous process, but other activities carried out by the procurement sector must not be the collateral damage of that process.

Such forecasts are more or less prone to significant errors, i.e., deviations in relation to later realized prices. There is no "best" form of forecasting in raw materials procurement, and there are few studies in the literature about forecasting conditions. Therefore, we sought to develop a model for the strategic procurement management of recycled paper in the paper industry by systematically applying strategic management principles.

Buyers/suppliers in raw material markets often apply ad hoc rules in forecasting and procurement, rarely testing long-term price series to properly assess their effectiveness. Most such rules are in fact impossible to test precisely because of their qualitative nature and inaccurate definitions. Most of these rules are designed to be used by speculators in markets whose only field of interest is whether prices will rise above or fall below current prices in the future. Kingsman et al. find that speculators have virtually no need to be in the market unless they are confident they will make a profit. However, a buyer from the industry has an objective need for raw materials, so it cannot avoid the market even in conditions when it is not able to make any real meaningful forecast. The buyer also wants to know how long prices will remain higher than the current prices, not just whether they will be higher at some point in the future [32,40]. Raw materials differ significantly from each other due to their physical properties, methods of production, places of origin, uses, distribution channels, etc. [41]. Such differences lead to specialization in raw material trade, with separate markets being developed for each commodity; an example is the renewable fiber market in the paper industry [37].

### 3. Research Methodology

The basic method of this research is a qualitative case study approach based on the primary and secondary data of four companies in the paper industry. A major reason for the popularity and relevance of theory building from case studies is that it is one of the

best (if not the best) of the bridges from rich qualitative evidence to mainstream deductive research. Its emphasis on developing constructs, measures, and testable theoretical propositions makes inductive case research consistent with the emphasis on testable theory within mainstream deductive research [42]. Multiple-case-study theory building recognizes theoretical patterns by systematically analyzing a small number of cases [43]. Its purpose is to develop novel, accurate, parsimonious, and robust theory that emerges from and is grounded in data. Case research is well-suited to address "big picture" theoretical gaps and dilemmas, particularly when existing theory is inadequate [44].

We applied a comparative multiple-case study according to the methodology of Yin [45]. While single-case studies can richly describe the existence of a phenomenon, multiple-case studies typically provide a stronger base for theory building [46]. We first collected secondary data, such as financial and strategic reports, presentations, and consulting materials, from four leading companies in the paper industry. In addition, primary data were collected through a survey administered to the procurement executives as well as via follow-up interviews with these executives. As research incorporates more cases and moves away from everyday phenomena such as work practices to intermittent and strategic phenomena such as acquisitions and strategic decision-making, interviews often become the primary data source. Interviews are a highly efficient way to gather rich, empirical data, especially when the phenomenon of interest is highly episodic and infrequent [42].

In the next step, we systematically organized all the obtained data for each of the individual companies in order to prepare all the necessary material for conducting a comparative analysis of the material collected from four companies. The aim of this comparative analysis was to identify key similarities and differences in the degree of maturity of the application of strategic management models across these companies, with a special focus on the application of these principles in the procurement of recycled paper.

### 3.1. Questionnaire Description and Validation

Respondents were experts in the paper industry, with a high level of education ranging from the fields of strategic management, economics, transport, logistics, and environmental management. The survey was meticulously designed in accordance with the research, the language area, and the corporate language of the selected companies (Appendix A). As can be seen from Appendix A, the structure of the survey envisages three sections of questions:

(1) Recycled paper procurement management;
(2) Recycled paper inventory management;
(3) Recycled paper procurement units.

A total of fourteen experts from the four sampled companies were interviewed using semi-structured interviews. The survey and interviews were conducted in late 2020 and during the first quarter of 2021. In constructing the survey, it was necessary to unite all the issues of this research and reduce them to concise and clear content. In that procedure, the previously collected data were sublimated and transposed to convey to the expert the problems of research within thirty open-ended and closed-ended questions. All experts were previously acquainted with the survey and their role in this research. The first section (recycled paper procurement management) addresses the specifics of forecasting price trends in the field of recycled paper in the paper industry. This section covers strategic management aspects in procurement and then the paper industry in general with a focus on key factors influencing the recycled paper planning process, then on strategic aspects related to the price of recycled paper in the global market (intense volatility, forecasting problem, recycled paper availability, market speculation) and strategic orientations of its procurement (procurement modalities, market appearance strategies, typologies of commercial arrangements). In addition, other disruptive factors and challenges to planning were considered, such as changes in end-customer requirements, complaints, high costs of recycled paper preparation, cooperation with production and the technical sector, etc.

The second part of the survey (recycled paper inventory management) is structured to consider all the specifics regarding the optimal management of inventory levels of

this raw material. All factors that directly and indirectly affect the dynamics of recycled paper movement through the warehouse are considered, such as changes in the production process, limited storage capacity, supplier influence, transport and logistics challenges, force majeure, and other factors. The frequencies of these phenomena were examined, as well as the existing strategies used by the studied companies in order to overcome the problem of mismatched stocks of recycled paper. Other factors are included, such as the correlation of low and high prices with the quality of incoming raw materials (indirect impact on inventory movements), new technologies for recycled paper preparation, optimization in the time-saving segment in the form of the automation of certain processes, and other technical specifics affecting recycling inventory management.

Finally, the last section of the survey covered questions related to the complexity of the organizational unit responsible for managing the procurement of recycled paper. Issues related to the position of the procurement function in the organization are included to assess the degree of maturity of the observed organization in this segment, whether its strategic importance in the organization is recognized. Therefore, the survey design aimed at adequately covering the issues of this research with thoughtful questions so that the answers of the experts could first be systematically organized and prepared for comparative analysis.

Interviews with experts were conducted in person (face to face) and online (using Skype). Interviews lasted 90 min on average, while with certain experts they lasted 120 min. The shortest interview was 60 min long. The duration of the interviews was similar to those of previous research and in line with best practices. All key research topics were discussed with the experts, following the agenda of the survey. Still, they were able to explain their views and considerations in more detail through a direct conversation. It was especially important that the experts were able to express their opinion on strategic developments in the paper industry and problems in the procurement of recycled paper and to analyze and discuss some open questions from the survey, which gave perceptions from each expert on research issues. In addition, each of the experts has their specialty, so the topics related to these fields of expertise were additionally communicated. As previously explained, the conducted survey and semi-structured interviews were anonymous, and accordingly the interviews were not recorded. Consent was obtained to record the interview through notes and to subsequently proceed with the transcription from which the obtained conclusions were drawn for this research.

### 3.2. Research Samples Description

The surveyed experts were selected based on the procurement organizational structure in the observed companies (see Table 1). As can be concluded from the table, four experts were sampled from the first two companies, while three others were selected from the remaining two companies. The number of experts was determined depending on the complexity of the recycled paper procurement sector of the observed companies. Based on these criteria, key managers with enviable experience, a high degree of professional qualifications, and autonomy in decision-making within their organization were meticulously selected.

**Table 1.** Structure of experts and their fields of expertise.

|  | Company | Fields of Expertise | Experience in the Paper Industry |
|---|---|---|---|
| Expert 1 | number 1 | 3, 4, 8 | 20+ years; |
| Expert 2 | number 1 | 6, 7, 10 | 6–10 years; |
| Expert 3 | number 1 | 4 | 0–5 years; |
| Expert 4 | number 1 | 6, 7, 10 | 6–10 years; |
| Expert 5 | number 2 | 1, 2, 8 | 11–15 years; |
| Expert 6 | number 2 | 1, 2, 3, 4, 6, 8 | 11–15 years; |
| Expert 7 | number 2 | 4, 6, 7, 10 | 6–10 years; |
| Expert 8 | number 2 | 1, 2, 5, 8 | 16–20 years; |

**Table 1.** *Cont.*

|  | Company | Fields of Expertise | Experience in the Paper Industry |
|---|---|---|---|
| Expert 9 | number 3 | 2, 4, 6, 7 | 16–20 years; |
| Expert 10 | number 3 | 1, 2, 4, 6, 7, 9 | 11–15 years; |
| Expert 11 | number 3 | 1, 6, 7, 10 | 20+ years; |
| Expert 12 | number 4 | 4, 6, 7 | 16–20 years; |
| Expert 13 | number 4 | 4, 6, 7 | 20+ years; |
| Expert 14 | number 4 | 2, 4, 6, 7 | 11–15 years; |

Legend:
Fields of expertise

1. Strategic management;
2. Economy;
3. Process management in the process industry;
4. Packaging waste management;
5. Market of pulp and finished paper products;
6. Market of recycled paper and other secondary raw materials;
7. Management of recycling paper collection centers;
8. Management of procurement of basic raw materials;
9. Environmental management;
10. Transport and logistics.

Experience in the paper industry:

- 0–5 years;
- 6–10 years;
- 11–15 years;
- 16–20 years;
- 20+ years.

## 4. Results

### 4.1. Review of the Results of Comparative Analysis

The procurement of raw material has strategic importance for production-oriented firms [8]. Therefore, forming an organizational unit that oversees the raw material procurement management process entails great responsibility. To successfully implement corporate policy in this part of the firm, it is necessary to design and adjust the organizational structure in accordance with the characteristics of raw materials and production in the firm.

Although the range of values (Table 2) was from 1 to 5, it can be seen that the obtained values are quite similar for all four companies and gravitate towards the value of 2. It turns out that experts believed that among the competitive advantages in the paper industry, there is not much room for giving higher priorities to one over the other. It is interesting that at the top of the list (in total/on average) is the strength of the sales network of final paper products and the strength of the procurement/collection network is in third place, although experts mostly occupied procurement-related positions.

**Table 2.** Comparative overview of results—competitive advantage in the paper industry.

|  | Summary Results | | | | |
|---|---|---|---|---|---|
|  | P1 | P2 | P3 | P4 | Avg. |
| Strength of sales network/placement of final products | 2.00 | 2.50 | 1.33 | 2.33 | 2.04 |
| Growth (strength) in production (recycling) capacities | 2.75 | 2.25 | 2.00 | 1.33 | 2.08 |
| Strength of procurement/collection network of basic raw materials (recycled paper) | 1.75 | 1.50 | 2.33 | 3.33 | 2.33 |
| Stable production work and technical support | 3.00 | 3.75 | 2.67 | 2.33 | 2.94 |

From the presented Table 3, the four companies relied on the method of price planning by collecting information from the market in more than 30% of cases. These data highlight the importance of quality market research and constant vigilance. Professionals in recycled paper procurement tend to listen to and follow developments in the finished products market by nature. This method is used automatically with the hope that in combination with other factors it will contribute better price forecasting.

**Table 3.** Comparative overview of results—methods for forecasting recycled paper price trends.

| Method | Summary Results | | | | |
|---|---|---|---|---|---|
| | P1 | P2 | P3 | P4 | Avg. |
| Price planning based on information obtained from the market (suppliers, customers, traders) | 33.75% | 33.75% | 36.67% | 33.33% | 34.38% |
| Price planning based on indices and professional publications (Euwid, Foex PIX, Risi) | 23.75% | 31.25% | 21.67% | 36.67% | 28.33% |
| Price planning based on your own intuition and analysis | 22.50% | 12.50% | 28.33% | 13.33% | 19.17% |
| Price planning based on price movements of finished paper products | 12.50% | 17.50% | 13.33% | 16.67% | 15.00% |

This segment was important to analyze because it is the starting point when contracting the procurement of recycled paper (Table 4). The experts unequivocally agreed that the above three determinants are the basis for deciding on recycled paper procurement. Moreover, the general message was that it is risky disrespectful to make a distinction regarding which of the determinants is more important than the others. However, viewed from a different angle, the production will certainly look for the best possible quality of input raw materials. This is because, in that way, it is easier to reach the required quality parameters of the finished product considering increasingly demanding end customers. In other words, the production does not have such an overview of market developments and that is why it is always necessary to find a balance between these two functions.

**Table 4.** Comparative overview of results—analysis of priorities of key determinants when contracting the procurement of recycled paper.

| | Summary Results | | | | |
|---|---|---|---|---|---|
| | P1 | P2 | P3 | P4 | Avg. |
| Quantity (raw materials availability) | 1.25 | 2.25 | 1.33 | 2.00 | 1.71 |
| Cost price | 2.25 | 1.25 | 1.67 | 2.00 | 1.79 |
| Raw material quality (fiber utilization) | 2.50 | 1.50 | 3.00 | 1.67 | 2.17 |

From Table 5, it can be concluded that the changing dynamics of the production process is the predominant factor of instability and disproportion of recycled paper stock in all four surveyed companies. All experts noted this issue as a disruptive factor in the process of inventory management.

**Table 5.** Comparative overview of results—analysis of key factors influencing the disproportion of recycled paper stocks.

| | Summary Results | | | | |
|---|---|---|---|---|---|
| | P1 | P2 | P3 | P4 | Avg. |
| Variable dynamics of the production process | 31.25% | 26.25% | 30.00% | 38.33% | 31.46% |
| Supplier unreliability | 14.75% | 27.50% | 20.00% | 16.67% | 19.73% |
| Limited storage capacity in the factory | 17.50% | 17.50% | 25.00% | 16.67% | 19.17% |
| Limited storage capacity in the factory | 17.50% | 17.50% | 25.00% | 16.67% | 19.17% |
| Influence of force majeure | 5.25% | 8.75% | 8.33% | 6.67% | 7.25% |

## 4.2. Model

The results show that the paper industry is mature enough to consider a strategic model to improve recycled paper price trend planning and contribute to a better approach to dealing with a dynamic market. Based on the results of a comparative analysis of the cases of four selected companies and a literature review on procurement in the paper industry, we propose a new model (Figure 1) for strategic procurement that contains three main building blocks, as explained below.

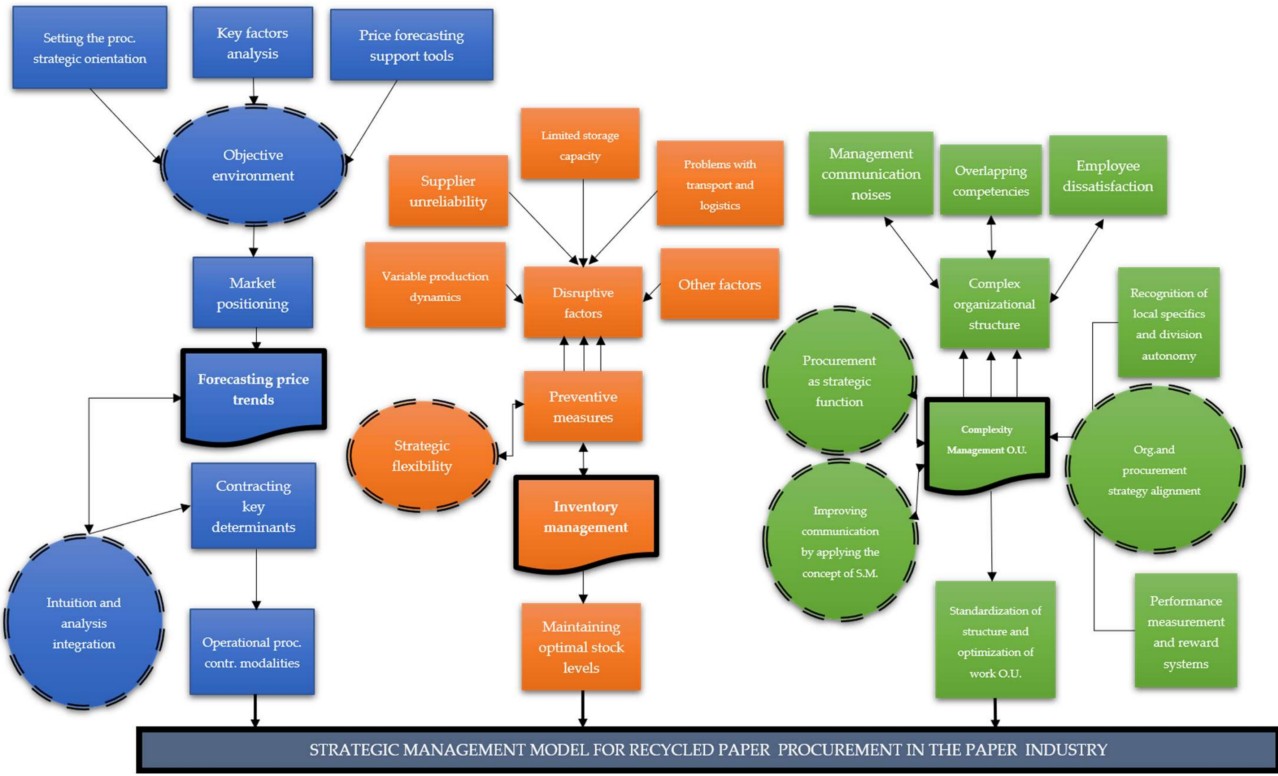

**Figure 1.** Proposed recycled paper procurement model for the paper industry.

The structure of the proposed model includes three main building blocks (black lines):

(1)     Forecasting price trends;
(2)     Inventory management;
(3)     Organizational unit management.

Additionally, the circular elements framed by dashed lines represent principles for the strategic management of the procurement of recycled paper. As can be seen from the model, the units are integrated into the whole at the end of their algorithmic construction due to them being interdependent processes.

The algorithm begins with elements related to the determining preconditions for the objective environment. Namely, the results indicate that there is an impact on the objective environment and related uncertainty. Although the collective actions of others affect the environment for a particular organization, they do not represent the integrity of an objective environment that includes both social and physical phenomena. However, the model recognizes that the collective actions of others determine the extent to which strategic and structural decisions, as well as performance, affect the objective environment and its uncertainty. Therefore, the setting of the procurement strategic orientation determines the way a firm will conceptually approach key raw material procurement.

In the paper industry, the procurement of recycled paper is undoubtedly a top priority due to the challenges of finding significant quantities on the dynamic market of a prescribed quality. It can be achieved through the development of its own collection network where the firm establishes direct access to generators (primary sources) of recycled paper or

through a policy of not investing in its own collection network but through a system of contracting with large and/or smaller collectors. This determines what the environment of the organization will look like and who the participants will be. More than 70% of the surveyed experts agreed on the position that they should own and develop their collection network and have access to generators (primary sources) of recycled paper and thus provide a better long-term strategic commitment. Then, the analysis of key factors in the paper industry implies active monitoring of the action of these factors, having in mind their strength, dynamism, and the possibility that in a certain period some weaken while others strengthen, which changes the direction of the industry.

These factors represent the business environment creators in the paper industry, and therefore their constant monitoring is important. In addition, it is necessary to include tools to support price forecasting, the use of which can provide certain indications of price trends. However, tools and methods (such as, price planning based on market information, indexes and professional publications, price movements of finished products, etc.) have a degree of uncertainty and serve as a possible indicator of expected price trends whose actual outcome is not guaranteed. After setting these three elements, their information output is processed by the principle of objective environment recognition. This principle is part of the strategic model according to Jauch and Kraft, who set the revisionist model of uncertainty in the environment, which assumes the desirability of uncertainty, while insisting on the importance of an objective environment [47].

According to Chambers and Quiggin, uncertainty is defined as a state of using limited knowledge to accurately describe an existing state or future outcome. It is measured by the assigned probabilities of each possible state or outcome [48]. Merigo introduced a more recent approach in dealing with risk and uncertain environments. He has developed the decision-making application by using a multi-person analysis that permits us to consider the opinion of several persons in the analysis. He suggests using this model in strategic management to deal with environments affected by both risk and uncertain factors and to assist the decision maker to make the most efficient decision [49]. On the other hand, Hong et al. claim that procurement in uncertainty or risk is predominantly reliant on the manager's experience and intuition, as there is no systematic way for them to classify the source of procurement risk, and a myopic risk management strategy is formulated once the unexpected risk has occurred. They claim that procurement risks can include factors such as price uncertainty (or risk, volatility), demand uncertainty (or risk), yield uncertainty (or risk, random), lead time uncertainty (or risk, variability), and disruption risks [50].

This phase of the model is the most challenging because it is based on all the information necessary to reach an objective understanding of the environment using the described principles. Based on the given knowledge of the objective environment, a position is taken on the market from which price forecasting is performed. Taking a position on the market depends on objective knowledge of the environment, so from correctly set positions, price trends can be more accurately predicted and more realistic market expectations can be gained.

After that, we come to the second strategic principle in this segment of the model (adopted from the Fred and Forest David model), which is the principle of using the integration of intuition and analysis in the procurement phase (elements in the model: key determinants in contracting and operational procurement modalities). This principle is especially good for making important decisions in situations of great uncertainty, which is an adequate qualification for recycled paper procurement within its market environment dynamic. According to this model, it was pointed out that intuition and rational process play a crucial role in effective strategic decision-making, which is especially important for the last two phases. This principle is assigned to this part because it is especially useful when it is necessary to choose between several credible alternatives, such as deciding on the choice of offers of several comparable suppliers. This part is the result of the price forecasting process, which concludes this unit and connects it with the main process of the model, which continues to the other two.

Managing the optimal level of recycled paper stocks is represented in the second logical unit of this model and starts with the factors that disrupt this process, among which, based on our research, the most important factor disrupting the optimal stock is the variable production dynamics factor. Accordingly, the strategic flexibility principle (adopted from the model of T. Wheelen and D. Hanger) is included to contribute to their efficiency. The basic activities of this principle are:

(1) Systematic problem solving;
(2) Experimenting with new approaches;
(3) Learning from one's own experience and from the experience of others;
(4) Fast and efficient transfer of knowledge through the organization that should be implemented to resolve internal disruptive factors (such as conflict between raw material procurement and production).

The strategic flexibility principle advocates the creation of a learning organization, which is especially important for inventory management problems because of the similarity of problems, and this principle is very effective for overcoming them. The same principle applies to external factors, such as unreliable suppliers or problems with transport and logistics, where it is necessary to persistently look for alternative solutions and try new approaches. With the described approach to these problems in combination with the existing preventive measures, a better outcome is expected in maintaining the optimal level of stocks, which is the goal of this unit.

The last part of the model is intended for organizational units managing the complexity of recycled paper procurement. It starts with the key elements that result from a complex organization: communication noise, overlapping responsibilities, and employee dissatisfaction. Within the management of this problem, there are three sets of principles of strategic management:

(1) The principle of recognizing procurement as a strategic organizational function;
(2) Communication improvement by applying the concept of SM;
(3) Harmonization of the organization's strategy with the raw material procurement function strategy.

One of the preconditions for dealing with this problem in general is that the organization recognizes raw material procurement as a strategic function. However, based on the results of the comparative analysis, the procurement function is still often underestimated in modern business. In addition, communication improvement, especially cross-sectoral communication in complex/divisional structures, using the concept of strategic management (according to Fred and Forest David) is another strategic principle within this unit. Alignment of the organization's strategy with the raw material procurement strategy is also one of the key steps that should be taken to prevent conflicts and achieve good results. The principle of closing the loop between the business strategy of the organization and the functional strategy is the unavoidable approach of organizations in this industry due to the elements of recognizing local specifics, division autonomy, and the existence of performance measurement and reward systems being directly linked. At the end of the third part, there is a work structure and optimization standardization of the organizational unit, which includes control of all processes and reducing raw material procurement to a rational level for all efficient processes. This concludes the third unit which, following the example of the previous two, constitutes the main process behind the model.

The model is designed in such a way that it can be applied in various firms in the paper industry, regardless of the characteristics (e.g., size) of the firm.

## 5. Discussion

This research began by noting the challenges in the existing procurement management models in the paper industry. The main contribution of our research is in developing a novel, strategic model for procurement management in the paper industry that helps professionals in this industry to manage the procurement process in a more efficient and effective way.

The development of the model was based on the integration of insights offered by the literature review, review of specialized consulting reports and professional publications dedicated to recycling paper and, most importantly, the collection and analysis of primary and secondary data on procurement processes in four firms in the paper industry.

The model we propose is based upon a combination of theoretical insights and best practices. Observed from the angle of recycled paper procurement strategic orientation, the research considered two options from the perspective of the organization:

1.  Owning and developing its collection network and having access to recycled paper generators (primary sources);
2.  Concluding commercial and spotting contracts with large and small collectors without collection network investments.

The consideration of these options is essential for the concept of procurement planning and forecasting price trends. Which strategic commitment a given organization decides upon depends on its conception of recycled paper procurement. In conversations with experts, more than 70% of them were of the opinion that the right strategic commitment, in the long run, is developing their collection network. Other experts were of the opinion that the combined application of both options may be a better solution. It follows from the above that the recycled paper stocks problem is too complex for us to find a unique solution that would include all aspects and challenges in recycled paper stock management. Therefore, as mentioned, several methods should be used in an adequate combination to achieve good results in this segment. As described, production is a "living thing" and its oscillations are difficult to predict. On the other hand, this is significantly reflected in the unit responsible for the procurement of recycled paper and its performance in the management of the optimal level of inventory and, therefore, in other aspects that were previously described. Consequently, good cooperation in the field of production and procurement is of great importance and this reflects on the entire organization [51].

Based on our results, we propose a model for recycled paper procurement in the paper industry that is based upon strategic management principles and includes three interrelated logical units:

a.  Forecasting price trends;
b.  Inventory management;
c.  Raw material procurement unit complexity management.

Analyzing the literature in the paper industry and based on expert knowledge obtained from a conducted multiple-case study, we found that the recycled paper market is uncertain and indirectly influenced by the described factors, leading to frequent and sudden changes in pricing. In addition, the market abounds in a large number of participants, and each of them has a specific influence (economic, environmental, political, etc.), which further complicates the monitoring of price flows [52,53].

There are several novel insights that emerge from our research. Among the surveyed experts in the paper industry, there was a consensus that the application of strategic management principles in recycled paper procurement would contribute to the overall results of the organization. This contribution would come through the improvement of three key segments in recycled paper procurement—price forecasting, inventory optimization, and having an efficient unit in charge of procurement.

Figure 2 represents a comprehensive descriptive overview of the results of the conducted comparative analysis observed through three key procurement recycled paper segments (more precise price forecasting—supply optimization—efficient procurement unit). The starting points are factors that influence the industry and create new trends and needs. The market reacts to them and through continuous market competition. In order to achieve competitive advantages, organizations direct their activities towards strengthening their sales network, production, (recycling) capacity, and expanding their recycled paper collection network, etc. Based on these insights, priorities are specified in the form of quantity, price, quality, or other parameters. This segment was very important to determine

because it is the starting point for contracting recycled paper procurement. The examined experts unequivocally agreed that the above three determinants are the basis for deciding on recycled paper procurement. Additionally, the general consensus was that it is risky disrespectful to make a distinction regarding which of the determinants is more important than the others. However, viewed from different angles, the production will certainly demand the best possible quality of recycled paper input, because in that way it will be easier to reach the necessary parameters of the quality of the finished product in light of increasingly demanding end customers. On the other hand, production does not have such an overview of market developments and that is why it is always necessary to find a balance between these two sectors.

**Planning**

**Paper industry growth and development key factors**

| | |
|---|---|
| Global economic development | 1.81 |
| Development of internet technology | 2.21 |
| Applications of circular economy principles | 2.21 |
| Growth and development of industrial production | 2.38 |
| EU strategies | 2.46 |
| Reducing the use of plastic products | 2.56 |
| The trend of switching to renewable fibers | 2.60 |
| Force majeure | 2.83 |
| Key energy oscillation prices | 2.83 |
| Demographic development (global population growth) | 2.88 |
| Global political developments | 3.10 |

**Competitive advantages in the paper industry**

| | |
|---|---|
| The power of the sales network | 2.04 |
| production capacity | 2.08 |
| Production capacity growth | 2.23 |
| Stable production work and technical support | 2.94 |

**Priorities analysis in contracting the recycled paper procurement**

| | |
|---|---|
| Quantity (raw materials availability) | 1.71 |
| Purchase price | 1.79 |
| Raw material quality (fiber utilization) | 2.17 |

**Recycled paper price trends forcast methods**

| | |
|---|---|
| Price planning based on information obtained from the market | 34.38% |
| Price planning based on indexes and professional publications | 28.33% |
| Price planning based on your own intuition and analysis | 19.17% |
| Price planning based on price movements of finished paper products | 15.00% |

**Recycled paper procurement structure from the aspect of contracting model**

| | |
|---|---|
| By contract-bound index formula | 57.50% |
| Spot contracting on a monthly basis | 42.50% |

**Supplies**

**Analysis of key factors influencing the disproportion of recycled paper stocks**

| | |
|---|---|
| Variable dynamics of the production process | 31.46% |
| Supplier unreliability | 19.73% |
| Factory limited storage capacity | 19.17% |
| Transport and logistics problems | 17.40% |
| Force majeure influence | 7.25% |

**Preventive measures analysis aimed at more efficient recycled paper inventories**

| | |
|---|---|
| Forming security stocks | 1.77 |
| Forming internal agreements with production | 2.71 |
| Making a storage arrangement with a supplier | 2.94 |
| Creating more flexible arrangements with | 3.23 |
| Lease of external storage space | 3.50 |

**Advantages and disadvantages of recycled paper storage analysis**

+Advantage: Possibility of storing key qualities (i.e. brown and deinking, etc.) in the open.
+Advantage: Resistant to high and low storage temperatures.
+Advantage: Relatively easy to handle in storage (storage in bulk and baled form).
+Advantage: Does not require daily maintenance.
-Disadvantages: Increased risk of fire.
-Disadvantages: Deterioration of quality (i.e. recycled paper deterioration) after prolonged storage in the open.
-Disadvantages: Risk of injury to warehouse workers during operations (i.e. treatment of RP).

**Unit**

**Key values of the procurement function**

Recognizing the recycled paper procurement function as a strategic function in the organization, as one of the prerequisites to provide good results.

Alignment of the organization's strategy with the functional recycled paper procurement function is crucial in order to avoid a conflict of goals. Furthermore, it is essential that the procurement function supports the company's corporate strategy.

The scope of the recycled paper procurement function has a wide range of activities and responsibilities at all three levels of management (i.e. strategic, tactical and operational level).

In addition to commercial skills and economic knowledge, the manager of recycled paper procurement needs to have a solid level of technical knowledge (properties of recycled paper, key processes of production, paper recycling, etc.).

Performance measurement and reward systems are important for motivating all members of the recycled paper procurement unit team, and thus contribute to achieving good results of this function.

**Figure 2.** Summary of the results of the comparative analysis.

Raw material availability is a prerequisite for the other two (price and quality) factors. Therefore, when planning recycled paper procurement, one should start from these determinants, with a special focus on the objective consideration of the raw material availability, and then consider the price and quality. This segment is indirectly correlated with price trend forecasting because it is done through the prism of these three determinants (raw material availability → price range → quality level). Accordingly, these determinants are an integral part of the model within recycled paper price forecasting trends.

Our research confirmed that forecasting recycled paper price trends is one of the biggest challenges faced in raw material procurement, to which the first part of the model is dedicated. Existing tools for price forecasting are not reliable and full reliance on them is not recommended, with which the examined experts agree. Given the significant number of actors, both on the supply side and on the demand side, it is difficult to accept the view that prices are not formed on market principles. In addition, the process of price forecasting is conducted using methods in which equal account is taken of the structure by which the procurement will be realized (i.e., through contracts related to a particular index or procurement spot). The procurement planning and implementation process is directly related to inventory management.

Factors that hinder the optimal level of basic raw material stocks are detailed and special attention is given to the changing dynamics of the production process and the importance of cooperation with the production sector [12,54]. Existing preventive measures that should prevent stock disproportion were considered, but their effect was assessed by experts as insufficiently successful to be characterized as a systemic solution. The procedure of recycled paper storage is not complex but it requires the responsibility of competent persons, especially in the field of protection and safety at work. The responsibility for managing the above processes lies with the raw material procurement unit. The complexity and scope of work performed by this unit requires a high degree of training of its staff in various disciplines (economic, technical, social, etc.) [55,56]. Because of often complex organizational structures, there is a shortcoming in monitoring the work of this unit, which can lead to a decline in its efficiency. Therefore, it is necessary to have appropriate performance measurement and reward systems, whose role is to prevent these phenomena [57].

Finally, the alignment of the organization's strategy with the functional procurement strategy (recycled paper) is also one of the prerequisites for achieving good results, both in procurement and the organization as a whole [58,59]. Then, the scope of the raw material (i.e., recycled paper) procurement function is present at all three levels of management, which only supports the importance of this function and its broad operation [60]. Executives managing raw material procurement need a wide range of knowledge [40,51]. This is an important conclusion that testifies only to what a demanding position it is. Therefore, it is necessary to have adequate systems for measuring performance and rewarding all members of the team working in the raw materials (recycled paper) procurement unit because their joint effort is critical for achieving the set goals of this function.

## 6. Conclusions

The results of this study indicate that the function of raw material procurement is recognized as strategic by executives and is treated alongside other organizational units. However, based on the comparative analysis results, we can see that in some situations this function is underestimated in relation to others. This fact supports the need for a model that strategically deals with the issues of raw material procurement. The results also indicate the importance of having adequate remuneration systems for the members of the unit responsible for the raw material procurement, especially considering the difficulty of the work they perform. The model we propose should help manage the complexity of the organizational unit for raw material procurement.

The main reason for the limited application of the principles of strategic management in the procurement of raw materials in the paper industry is likely the scarce research conducted in this field. The deficit of recent research is particularly pronounced, which

results in the absence of the application of modern strategic models in practical terms, which further increases the importance of this research. These challenges were the motive for conducting the current study, which resulted in a comprehensive model based on the synergy between the theoretical basis of the strategic management principles and practical inputs derived from real businesses in the paper industry.

The results of this paper must be seen in light of its limitations. A comparative multiple-case study was conducted in four industry-leading companies. The paper industry has more companies that are important participants in this market. Expanding the number of companies and experts would provide broader knowledge about the researched topics and strengthen critical thinking about certain considerations. Testing the model in companies as well as analyzing the results of its application are the main recommendations for future research. In addition, the author's recommendation for future researchers is to test the model in a number of companies with different structures and market orientations, preferably in different countries, to examine its application. Feedback from companies that have tested the model should be analyzed to consider whether it is necessary to make corrections in some areas.

Our research also opens up several important avenues for future research. Considering the similarity of branches in the process industry, the model can be modified according to the specifics of the given branches, which in their structure have a production process based on the use of raw materials with a separate organizational unit in charge of its procurement. In order to adapt the model and use it to its full potential, it is necessary to conduct data collection and research in the field of the given branch. We recommend applying the same research methodology (comparative case study analysis) to identify the key challenges faced by professionals in the procurement of raw materials. Another important avenue for future research is in testing the model in practice. On the basis of such an endeavor, future research could be conducted to investigate the firm performance implications of the adoption of our model. Finally, future studies could also examine the adoption rates as well as the diffusion of the model across firms in the paper industry.

**Author Contributions:** All authors contributed equally to this work. All authors have read and agreed to the published version of the manuscript.

**Funding:** This research received no external funding.

**Institutional Review Board Statement:** This study was omitted from ethical review and approval due to the anonymity of the respondents. By completing the survey and participating in semi-structured interviews, the respondents gave informed consent to the anonymous processing of their opinions.

**Informed Consent Statement:** Informed consent was obtained from all entities involved in the survey and semi-structured interviews opinion.

**Data Availability Statement:** All subjects gave their informed consent to inclusion prior to participation in the study. The study was conducted in accordance with the Declaration of Helsinki, and the research was conducted in accordance with the Code of Ethics of the University of Belgrade.

**Conflicts of Interest:** The authors declare no conflict of interest.

## Appendix A

Respondents assigned a value on a scale of 1 to 5 (1—highest value, 2—high value, 3—medium value, 4—low value, 5—lowest value) or influence on a scale of 1 to 5 (1 most significant, 2—very significant, 3—moderately significant, 4—slightly significant, 5—least significant) in accordance with the statements. Additionally, the respondents assigned percentages and other values in certain segments and had the opportunity to answer closed and open questions. The questions were in three sections:

(1)  Recycling paper procurement management:

   a.  Assess the impact of factors on changes in the paper industry.

b.   Evaluate the competitive advantages offered in terms of importance.

c.   Evaluate methods of recycled paper price trends forecasting in accordance with their use in the organization.

d.   Assess the importance of quality (fiber utilization), quantity (raw material availability), and purchase price when contracting procurement.

e.   Assess the importance of the proposed procurement management challenges.

(2)   Recycling paper inventory management:

a.   Estimate the value of the proposed factors that affect the fluctuation (disproportion) of inventory.

b.   Estimate the monthly frequency of procurement adjustments.

c.   Assess the importance of the proposed preventive measures in stockpile management.

d.   Assess the importance of automating the recycling of recycled paper.

(3)   Unit in charge of procurement of recycled paper:

a.   Assess the level of stress of the person employed in the position of Recycling Paper Purchasing Manager.

b.   Assess agreement with the statement that the procurement function is a strategic function.

c.   Evaluate the relationship between the procurement function and the organization's corporate strategy.

d.   Assess the connection of the procurement function with all three managerial levels in the organization (strategic, tactical, and operational level).

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
