# Peer review of "Strategic Turnaround in the Paper Industry: A New Model for the Procurement of Recycled Paper"

_sustainability, doi:10.3390/su14031475_

Round 1
Reviewer 1 Report
The statistical treatment of the study data is described in a rather superficial and ambiguous way, so it is not very clear how they were treated and the information that they managed to report to the study.
The results of the a priori study seem to be congruent with the objectives set out in the article, although it is no less true that they are described in an excessively brief and superficial way. Despite this, they conform to the methodological approaches established and planned in the study and are, for the most part, conveniently organized and written, at least from a chronological and statistical point of view.
The conclusions of the study are correctly stated, organized and described. However, and despite this, they are unable to clearly delimit the scientific space through which the different empirical studies that, in the future, intend to continue with the trail or the path that they have stopped exploring will have to continue or run the present study.
The references are quite current, despite the fact that in some cases they end up being older than ten years, and are reflected in the text of the article, scrupulously respecting the APA regulations in its seventh edition.
Reviewer 2 Report
This manuscript does an excellent job proposing a new model for the procurement of basic raw material, which helps increase the accuracy of forecasting price trends. This is a good case study and balanced assessment of the strategic turnaround in the paper industry. Only 1 area needs revision.
Page 13. Figure 2
Authors should consider simplifying this future. It is better to demonstrate analysis instead of data results here.
Reviewer 3 Report
Dear Authors, this is a good initiative to understand the extend to which strategic management principles/models are applied in the paper industry. Some comments for improvement as below:
- Abstract is not directly clear on the main issues of this article. It could be rewritten overall to emphasize main research issues, investigation and findings. line 10 states principles of strategic management, please specify which ones are you covering in your paper.
- some inconsistencies are detected in sentence/paragraph structures. e.g. line 73, the sudden discussion on strategic project management in this paragraph, lacks flow with the earlier part of the paragraph. found many such instances throughout this paper. E.g. line 108 on the sudden inclusion on planning.
- line 88 - controlled firme? refers to?
- Yin's methodology (2002) could be better supplemented with more recent approaches to comparative case study methodology
- What exactly of the topic/dimensions on strategic management that you would like to highlight/include in your research doesn't come clear from the beginning.
- should the study be clear from the beginning that it focuses on recycled paper procurement as the raw material input? the discussions that ensues seem to highlight this so why not make it clear in the title?
- To strengthen your argument/case for this study, you could show evidence from previous current studies on the value of implementing strategic management principles and practices to procurement. this is lacking in this manuscript.
- section 3.1. it is unclear he basis of these questionnaire? how were these statements formulated, designed, validated? self-developed or taken from previously tested sources?
- Table 1, could be reformatted to be consistent. Column 1 and other columns are not aligned.
- It is intriguing to me to notice Expert 6 from Company #2, has so many fields of expertise though the individual's experience in the paper industry is only 0-5 years? how is this justified?
- line 436, Jauh or Jauch?
- Jauch & Kraft [34], is a good work from 1986; however you may need to supplement this study with a more current discussions on this as you are discussing about procuring recycled raw materials for paper which are alot more current in my view.
- line 416, what does it mean by 'challenging qualitative?'
- at the end of the Discussions, I am still left wondering what is the main contribution of this study in applying strategic management functions in procurement.
- Figure 2 is a good attempt to summarize the descriptive analysis of this research. I see it as informative, but the linkage of one factor to another using arrow could infer a formative direction of relationship among the variables (which is not the case). e.g. Paper industry grown and development key factors 'do not lead to' competitive advantage in the paper industry... and so on. hence having these arrows seems like a misrepresentation of the descriptive data.
- Overall the references in this manuscript suffers lack of current research. A lot of work has been done in recent years in the field of procurement + strategic management, which can be included in the manuscript.
- spelling/grammar errors to a large extent that should be corrected, among them are found in: line 14, 16, 29, 88, 136, 157, 159, 212, 288, 292, 390, 392, 410,
Round 2
Reviewer 3 Report
Thank you for carrying out the corrections thoroughly.